# MRI-Based Radiomics for Outcome Stratification in Pediatric Osteosarcoma [note 1]

**DOI:** 10.3390/cancers17152586

**Published:** 2025-08-06

**Authors:** Esther Ngan, Dolores Mullikin, Ashok J. Theruvath, Ananth V. Annapragada, Ketan B. Ghaghada, Andras A. Heczey, Zbigniew A. Starosolski

**Affiliations:** 1Department of Radiology, Baylor College of Medicine, Houston, TX 77030, USA; 2Mary Bridge Children’s Hospital, Tacoma, WA 98403, USA; 3Department of Radiology, Texas Children’s Hospital, Mark A. Wallace Tower, 6701 Fannin Street, Suite 450, Houston, TX 77030, USA; 4Department of Pediatrics-Oncology, Baylor College of Medicine, Houston, TX 77030, USA

**Keywords:** machine learning, pediatric, Osteosarcoma, MRI, radiomics, classification

## Abstract

Osteosarcoma (OS) is a rare and aggressive bone cancer affecting children and adolescents and is associated with a low survival rate. Predicting disease progress or response to treatment is challenging due to the tumor’s complexity. This research aims to use advanced MRI techniques and machine learning methods to predict important outcomes including progressive disease, therapy response, relapse, and survival in pediatric OS patients. The findings show that these models can be highly accurate, enabling better decisions about treatment outcomes. This could lead to better risk stratification, treatment planning, and improving patient outcomes. The research also highlights the potential for these methods to be applied in other hospitals and research settings, benefiting the broader medical community.

## 1. Introduction

Osteosarcoma (OS) is the most common malignant bone tumor in children and adolescents. In the United States, approximately 1000 new cases of osteosarcoma are diagnosed each year, with about half of them occurring in children and adolescents [1]. OS originates when bone-forming cells become cancerous. OS is an aggressive form of bone cancer that can affect both bone and surrounding soft tissues. While the tumor primarily originates in the bone, it can invade nearby soft tissues, contributing to the disease’s progression and complicating treatment outcomes. OS most commonly develops in the ends of long bones, particularly around the knee. OS primarily occurs in the legs and arms, but it can also develop in other parts, such as the pelvis, shoulder, and skull [1].

The standard of care typically involves a combination of neoadjuvant chemotherapy, surgery, and adjuvant therapy to target any remaining cancer cells [1]. Radiation therapy might also be used, particularly when the cancer has spread to areas where surgery is not an option. Surgical options include limb salvage or amputation. Both of these treatment options can significantly impact the patient’s mobility and quality of life. Despite advancements in treatments including surgery and chemotherapy, outcomes remain suboptimal for certain patients. Specifically, survival rates drop significantly in cases of progressive disease during therapy, poor response to treatment, and relapse occurrence. The 5-year survival rate for localized OS is 64–76%, whereas for metastatic OS the survival rate drops significantly to 24% [2]. Diagnostic imaging plays a crucial role in the management of osteosarcoma, with X-rays, MRIs, and CT scans being commonly used. MRI is particularly valuable for providing detailed 3D images of both bone and soft tissue structures due to its superior soft-tissue contrast and non-ionizing nature [3], allowing for a precise assessment of the tumor’s extent. CT scans are primarily used to detect pulmonary metastases, as the lungs are the most common site of distant spread in OS patients [4,5].

Pediatric OS presents unique challenges compared to adult cases, as children and adolescents are still undergoing skeletal growth. This biological complexity, coupled with the heterogeneous nature of OS tumors, complicates the accurate prediction of key clinical outcomes. Existing prognostic factors such as tumor size, location, and histological response to chemotherapy provide some guidance but remain insufficient for precise risk stratification. There is a critical need for more advanced, non-invasive tools to improve the prediction of disease outcomes in pediatric OS.

Radiomics is an emerging field that involves extracting high-dimensional quantitative features from medical images, such as magnetic resonance imaging (MRI), to uncover patterns imperceptible to the human eye [6]. Radiomics data, when combined with other patient information, are analyzed using advanced bioinformatics tools to create mathematical models that have the potential to enhance diagnostic, prognostic, and predictive accuracy [6]. Radiomics has been applied to analyze different diseases, including cancers [7,8,9,10,11,12]. By analyzing these radiomic features (RFs), researchers can capture tumor heterogeneity, which is believed to reflect underlying biological processes such as tumor aggressiveness and treatment response. Radiomics has shown promise in various cancers, including brain tumors [13,14,15,16], lung cancer [17,18,19], and sarcomas [20,21,22,23], for predicting patient outcomes and guiding personalized treatment strategies. However, there is a lack of standardization in RF extraction and reporting across the field. The Image Biomarker Standardization Initiative (IBSI) has proposed guidelines to ensure reproducibility and comparability across radiomic studies [24]. While radiomics is an emerging field in cancer research, only a few studies have explored its application in OS [10,25,26,27,28], not to mention the IBSI recommended ones.

In addition to RFs, our analysis also considered demographics and pre-treatment clinical features such as skip lesions, OS subtype (osteoblastic, chondroblastic, etc.), OS location, and laterality. To further improve our classification performance, we adopted a hierarchical model that included prior outcomes as predictors for subsequent outcomes. Specifically, progressive disease and therapy response were included when modeling relapse after therapy, whereas the previous three factors were included when modeling mortality. Previous studies have been carried out to focus on individual clinical outcomes such as therapy response [10,29], pulmonary metastasis [5], relapse [27], and mortality [25,28,30]. However, these studies have generally examined each outcome separately without considering the interdependencies between them. The hierarchical approach enables us to capture the interdependencies of these clinical events, providing a more comprehensive view of the disease.

Furthermore, most prior studies have relied on whole-tumor segmentation without considering regional variations within the tumor, which could offer more insight into the tumor’s heterogeneity [25,26]. To address this gap, we explored two additional distinct segmentation strategies: (1) whole-tumor segmentation (usual practice); (2) tumor sampling from the whole tumor to account for intratumoral heterogeneity; and (3) bone/soft tissue separation. To our knowledge, no study has comprehensively explored pediatric OS by focusing on multiple clinical outcomes, MRI radiomics, and accounting for tumor structural heterogeneity using seeks to address these gaps via a comprehensive approach to analyze MRI data from pediatric OS patients with extremity tumors. Through this innovative methodology that integrates radiomics, advanced segmentation strategies, and combinations of clinical features, our study aims to enhance our understanding of pediatric OS and improve the prediction of critical clinical outcomes. Identifying reliable imaging biomarkers could facilitate early risk stratification, personalize treatment plans, and ultimately improve patient outcomes. This work represents a novel contribution to the field by addressing multiple clinical outcomes simultaneously and utilizing standardized radiomics features in a pediatric population. Preliminary results from this study were previously presented at the Society for Pediatric Radiology Annual Meeting [31].

## 2. Materials and Methods

### 2.1. Patients’ Cohorts

The study was conducted with approval from our institutional review board (H-50282). We identified 131 patients from a tertiary children’s hospital between 2006 and 2022. To identify predictors and outcomes, we searched through the clinical notes in Epic using keywords such as “relapse”, “progression”, “skip lesion”, and “histology type”. Only those with a complete medical record, pre-treatment post-contrast T1-weighted MRIs, and OS located in upper and lower extremities were included (Figure 1). Ultimately, 63 patients were included in the analyses using whole-tumor/tumor-sampling segmentation. Twenty-six of them underwent bone/tissue segmentation. An additional nine patients with their pre-treatment scans taken outside our facility were used as external validation.

### 2.2. Evaluated Outcomes

This study evaluated four binary outcomes in patients diagnosed with OS. Figure 2 shows the timeline for data collection and outcome evaluation. The first outcome was “progressive disease”, which assessed whether the disease progressed during the neoadjuvant and adjuvant phase, as documented in the clinical notes. This included, but was not limited to, metastasis after diagnosis, tumor regrowth, and relapse during or at the end therapy. The second outcome focused on “response to therapy”, defined as an “adequate” or “poor” response based on the percentage of necrosis on histopathology, with a threshold of 90% necrosis for a favorable response. If a range of necrosis percentages were reported, the mean value would be used for analysis (e.g., a range of 10–20% was replaced by 15%). The third outcome, “relapse/recurrence off therapy,” refers to any recurrence that occurred after the patient had been declared as having no evidence of disease (NED) or completed all the treatments. Relapse off therapy was coded as no if patients relapsed during or at the end of therapy. However, they would be coded as having progressive disease. Finally, the fourth outcome was “OS related mortality”. Patients who died from causes unrelated to OS were coded as no.

### 2.3. Segmentation Method

The study employed three segmentation methods to delineate the tumor regions (Figure 3). The first method involved whole-tumor segmentation, where the entire tumor area was outlined to capture its complete extent. This approach might include adjacent normal tissues at the tumor’s edges. The second method utilized a tumor sampling approach based on the above whole-tumor mask, dividing the tumor into seven distinct, non-overlapping regions corresponding to each face: top, bottom, front, back, left, right, and middle. The final method involved segmenting between bone and soft tissue regions within the tumor. The bone/soft tissue segmentation was performed by a pediatric radiologist with nine years of experience. Whole-tumor segmentation was subsequently conducted by a postdoctoral researcher with two years of experience in medical image analysis, building upon the initial bone/soft tissue segmentations. Segmentation was performed in 3D Slicer (ver. 5.2.2), and analyses were conducted in Python (ver 3.11.4).

### 2.4. Data Standardization and Features Sets

Both local and external MRIs were standardized to isotropic spacing and image pixels were min–max normalized. We selected three types of features for classification analyses. First, only radiomic features recommended by IBSI with no filter use were analyzed (IBSI RFs, *n* = 107). Second, both IBSI RFs and features with filters were considered (all RFs, *n* = 107 + 1177). Filters included wavelet and Laplacian of Gaussian transformations. RFs were calculated for each segmented region. Thus, segmentations involving bone/soft tissue yielded 2× more RFs than those based on whole-tumor segmentation. Similarly, region-based sampling had 7× more RFs than whole-tumor segmentation. Finally, demographics and pre-treatment clinical features were included when relevant, including laterality, histology (e.g., osteoblastic subtype), tumor location, metastasis at diagnosis, and presence of a skip lesion. Clinical variables were meticulously reviewed and extracted from Epic charts by a pediatric hematologist–oncologist with nine years of experience and a trained personnel to ensure accuracy and consistency. In addition to pre-treatment clinical features, we adopted a hierarchical approach to training classifiers for each of the four outcomes. For progressive disease and therapy response outcomes, only pre-treatment clinical features were included as potential predictors. The relapse outcome classifiers incorporated pre-treatment clinical features, progressive disease, and percentage of necrosis, whereas the mortality outcome classifiers encompassed all previous features, i.e., pre-treatment clinical variables, progressive disease, percentage of necrosis, and relapse.

### 2.5. Machine Learning

The ML pipeline began with splitting the dataset of 63 patients (or 26 for bone/soft tissue segmentation) into 80% for training and 20% for testing. RF reduction was performed on the training set in two steps. First, correlation analysis was applied to remove the highly correlated RFs. With IBSI RFs (i.e., RFs without use of filters) only, a Spearman correlation of 0.9 was used as the cutoff. For features involving filter use, a Spearman correlation of 0.8 was used as the cutoff. The varied cutoffs were chosen pragmatically based on the total number of features and segmentation strategy, while balancing feature redundancy reduction and preserving sufficient information for model training. In analyses using IBSI RFs, a strict 0.9 cutoff overly limited the feature set and reduced model flexibility in the subsequent steps. In contrast, with all RFs, the number of features increased dramatically to thousands (>7000 with tumor sampling). Thus, we applied a stricter cutoff of 0.9 to effectively reduce redundancy and computational burden. In the correlation-based feature reduction, priority was given to IBSI RFs. For models incorporating clinical features, features were added to the reduced radiomic set. Neighborhood Component Analysis (NCA) was then applied to the updated feature set (whether composed of RFs alone or combined with clinical features), and all features were ranked by their relevance to the outcome. It is possible that the best-performing classifier might not include clinical features if they were not ranked highly enough for inclusion, even though they were part of the initial feature set. Following feature reduction, we applied 5-fold cross-validation for all models, except those using bone/soft tissue segmentation, where 3-fold CV was employed due to the limited number of cases. Linear and nonlinear classifiers with different parameter settings were evaluated, including logistic regression, K-nearest neighbor (KNN), linear discriminant analysis (LDA), support vector machine (SVM), random forest, naïve Bayes, ensembles, and multi-layer perceptron network (MLP). Classification metrics, including receiver operating characteristic area under the curve (ROC AUC), precision–recall area under the curve (PR AUC), accuracy, sensitivity, and specificity, were calculated. Classifier performance was primarily evaluated based on the validation ROC AUC to determine the optimal classifier(s) for each outcome, feature type, and segmentation method. The same training set was used for analyses involving whole-tumor and tumor-sampling segmentation. In total, we selected 72 top-performing classifiers across four outcomes, three segmentation methods, and different combinations of radiomic and clinical features. Machine learning analyses were performed using the Scikit-learn Python package on a Linux workstation equipped with an Intel(R) Core(TM) i9-9900K CPU @ 3.60 GHz (16 logical cores), 64 GB of RAM, and a single NVIDIA GeForce RTX 1080 Ti GPU with 11 GiB memory.

### 2.6. Statistical Analysis

The Kruskal Wallis test, Chi-square test, or Fisher exact test, depending on the need, were conducted to compare the distribution of predictors and outcomes among different cohorts, including the full sample of 63 patients, the subset of 26 of them for whom bone/soft tissue segmentation was also performed, and the 9 patients with external scans. All statistical analyses were performed using a significance level of 0.05.

## 3. Results

### 3.1. Descriptive Statistics

Table 1 shows the summary statistics for the three cohorts: the full sample (63 patients), a sub-cohort with bone/soft tissue segmentation (26 patients), and an external cohort (9 patients). Among the 63 patients, the mean age was 11.82 (SD 3.53), with a predominance of males (43; 68.25%), Caucasians (52; 82.54%), and non-Hispanics (32; 50.79%). OS mostly occurred in the femur (41; 65.08%). Osteoblastic subtype (50; 79.37%) was the most common histological classification. At the time of diagnosis, 13 (20.63%) patients had pulmonary metastases, and 8 (12.70%) exhibited skip lesions. In terms of outcomes, 17 (26.98%) patients had progressive disease during therapy. The mean percentage of necrosis on histopathology was 93%. A total of 16 (25.40%) patients experienced relapse after therapy; 19 (30.16%) died from OS-related complications. No significance differences were found between the three cohorts in any variables except laterality, where patients with external scans had a significantly lower percentage of OS on the right side.

### 3.2. Outcome Interdependencies

Progressive disease was significantly associated with therapy response (*p* < 0.0001) (Table 2). Among patients with progressive disease during therapy, 14 out of 17 (82.4%) had a poor therapy response, compared to 12 out of 46 (26.1%) among those without progressive disease. Relapse off therapy was not significantly associated with any prior outcomes (*p* > 0.05). OS-related mortality showed a highly significant association with progressive disease during therapy (*p* < 0.0001) and relapse off therapy (*p* = 0.009). Metastasis at diagnosis was significantly associated with the occurrence of relapse off therapy and OS-related mortality (*p* = 0.008 and 0.037). Chondroblastic subtype was also significantly more prevalent in the deceased group (*p* = 0.033).

### 3.3. Classification Results

Depending on the feature set and segmentation approach used, the training times for classifiers varied. On average, each outcome- and segmentation-specific analysis required 1.5 to 4.5 h to complete. The shortest runtimes were observed when using IBSI RFs combined with whole-tumor segmentation, while analyses including all RFs and tumor-sampling segmentation required the longest processing times.

#### 3.3.1. Progressive Disease

Table 3 shows the detailed classification results for progressive disease. A list of selected features can be found in Appendix A. Classifiers derived from bone/soft tissue segmentation demonstrated higher validation and testing performance in terms of AUC compared to those developed using whole-tumor or tumor-sampling segmentation methods. A tumor being located in the humerus consistently ranked as the top clinical feature in the classifiers. Comparatively, the classifiers without any RFs generally had poor validation and testing classification results.

For bone/soft tissue segmentation, all top-performing classifiers showed excellent validation performance, with ROC AUC above 0.94. With IBSI RFs, adding baseline clinical features did not improve testing performance. The ROC AUC reached a maximum of 0.88 with an LDA classifier and a tissue RF (original_glrlm_RunLengthNonUniformity_tissue). While using all RFs optimized the validation performance, it reduced the testing performance and required an additional 13 features.

For whole-tumor segmentation, the best-performing classifier in terms of validation ROC AUC was a random forest classifier with 13 filtered RFs and OS location in the humerus. The classifier achieved ROC AUC 0.92 ± 0.09, with a testing ROC AUC of 0.51. Adding clinical features to filtered RFs improved validation ROC AUC, with the largest relative gain of 35.3%. However, the testing ROC AUC remained similar, with the highest value of 0.67 obtained without clinical features.

For tumor sampling, classifiers using all RFs slightly outperformed those using IBSI RFs in terms of validation ROC AUC. Compared to the classifier using IBSI RFs alone, adding baseline clinical features improved the validation ROC AUC to 0.84 (+12%) while decreasing the testing ROC AUC to 0.63 (−8.7%). Using only all RFs further increased the validation ROC AUC to 0.91 and the testing ROC AUC to 0.86 with an MLP classier and 15 RFs with filters (4 left, 4 front, 2 back, 5 bottom regions). Adding clinical features slightly reduced the validation ROC AUC to 0.90 but improved the testing ROC AUC to 0.89 (+3.5%).

#### 3.3.2. Response to Therapy

Table 4 shows the detailed classification results for response to therapy. A list of selected features can be found in Appendix A. In general, classifiers derived from bone/soft tissue segmentation demonstrated higher validation AUC values compared to those using whole-tumor or tumor-sampling segmentation methods. A chondroblastic subtype often ranked as one of the most important clinical features. Comparatively, the classifiers without any RFs generally had lower validation performance in terms of ROC AUC. Comparatively, the classifiers without any RFs usually had poor validation and testing classification results.

For bone/soft tissue segmentation, all top-performing classifiers achieved perfect validation classification results. However, on the testing set, adding clinical features led to decreased ROC AUC (max −16.7%). The best classifier overall belonged to the KNN (k = 5) model with four bone RFs with filters, which yielded perfect validation and testing classification metrics.

For whole-tumor segmentation, classifiers using all RFs slightly outperformed those based on IBSI RFs, with validation ROC AUC improving by up to 0.10 (+12.0%). Among models using IBSI RFs, an MLP classifier incorporating six RFs, chondroblastic subtype, gender, and presence of skip lesion, performed the best, with validation and testing ROC AUCs of 0.86 ± 0.09 and 0.80, respectively. With all RFs, an SVM with 14 RFs with filters had validation and testing ROC AUCs of 0.93 ± 0.06 and 0.78, respectively.

For tumor sampling, nearly all top-performing classifiers achieved a validation ROC AUC above 0.90, with the highest (0.98 ± 0.02) obtained from a polynomial SVM and 11 RFs. However, that classifier only had a testing ROC AUC of 0.6. The second-best classifier belonged to a random forest classifier using three IBSI RFs (top, left, right region), subtype OS, and gender, yielding a validation ROC AUC of 0.94 ± 0.05 and the highest testing ROC AUC of 0.76.

#### 3.3.3. Relapse off Therapy

Table 5 shows the detailed classification results for relapse off therapy. A list of selected features can be found in Appendix A. Classifiers derived from bone/soft tissue segmentation demonstrated higher validation AUC values compared to those based on whole-tumor or tumor-sampling segmentation methods. Among demographic and clinical features, the presence of skip lesions and metastasis at diagnosis consistently ranked among the most important predictors. In contrast, adding prior outcomes (i.e., progressive disease and percentage of necrosis) did not improve classification performance.

For bone/soft tissue segmentation, all top-performing classifiers achieved perfect validation classification results, except for the model using only IBSI RFs. Models using all RFs achieved comparable validation and testing results relative to the IBSI RF models. However, this required 3× as many features and had slightly lower testing accuracy and specificity. The best-performing classifier was an SVM model using three features (skip lesion, original_glszm_ZoneVariance_tissue, original_firstorder_Energy_bone), with a testing ROC AUC and PR AUC of 0.75.

In the case of whole-tumor segmentation, the highest validation ROC AUC (0.89 ± 0.11) was obtained by an SVM classifier incorporating eight filtered RFs, along with metastasis at diagnosis and skip lesion. However, the model’s testing ROC AUC was only 0.56. In comparison, an MLP using five features (metastasis at diagnosis, skip lesion, and three filtered RFs) achieved a validation ROC AUC of 0.78 ± 0.25, a testing ROC AUC of 0.73, and perfect sensitivity (1.0).

In tumor sampling, classifiers using all RFs showed marginal improvements in validation performance and similar test ROC AUCs, consistently yielding higher testing sensitivity at the cost of reduced accuracy. An MLP classifier with ten features (metastasis at diagnosis, skip lesion, and eight filtered RFs) attained the highest validation ROC AUC (0.93 ± 0.07), though its testing ROC AUC was 0.67 with perfect sensitivity. Using IBSI RFs, a polynomial SVM with 13 features (metastasis at diagnosis, skip lesion, chondroblastic subtype, humerus location, and nine RFs) achieved a validation ROC AUC of 0.86 ± 0.13, a testing ROC AUC of 0.73, and a specificity of 0.90.

#### 3.3.4. OS-Related Mortality

Table 6 shows the detailed classification results for OS-related mortality. A list of selected features can be found in Appendix A. Most classifiers derived from bone/soft tissue segmentation achieved perfect classification performance on both the validation and testing sets, outperforming those based on the other two segmentation approaches. These high-performing models included a random forest classifier incorporating progressive disease, three filtered RFs (2 bone RFs, 1 tissue RF), and a sigmoid SVM with seven IBSI features (four tissue RFs, three bone RFs).

For whole-tumor segmentation, classifiers using filtered RFs generally outperformed those based on IBSI RFs (ROC AUC max + 44.9%). The best-performing model was a KNN classifier (k = 5) using six features: progressive disease, relapse status, percentage of necrosis, and three filtered RFs. This model achieved perfect validation performance, a testing ROC AUC of 0.97, and perfect classification accuracy on the external test set. Incorporating prior clinical outcomes significantly improved the performance of classifiers using IBSI RFs but had limited benefit when all RFs were included.

With tumor-sampling segmentation, the top-performing model in terms of validation ROC AUC was an SVM classifier using only three prior outcomes: progressive disease, percentage of necrosis, and relapse status. This model achieved a validation ROC AUC of 0.98 ± 0.03 and a testing ROC AUC of 0.92. The second-best model was a KNN (k = 8) classifier with eight features (progressive disease, relapse status, six filtered RFs), achieving a validation ROC AUC of 0.98 ± 0.04 and the highest testing ROC AUC of 0.94. This model also obtained an accuracy of 0.78 on the external set.

Figure 4 shows the middle slice of an MRI from patients who were either misclassified or correctly classified by the majority of the models. The patients’ demographics and clinical information are presented in Table 7.

## 4. Discussion

This study explores the use of filtered and IBSI-validated radiomics using MRI data from pediatric OS patients with extremity tumors. Our models demonstrated robust predictive capabilities for multiple clinical outcomes, including progressive disease, therapy response, relapse occurrence, and mortality. The hierarchical approach, which considered prior outcomes as predictors for subsequent events, captured the interdependencies of clinical events and provided a more holistic view of individual outcomes. By integrating clinical features and hierarchical modeling, we captured the interdependencies of these clinical events, providing a more comprehensive view of disease trajectories. Certain combinations of features and classifiers yielded good generalizability to external data, especially the ones for predicting OS-related mortality, demonstrating the potential of our models to be applied across different healthcare facilities and datasets. This generalizability underscores the practical utility of our findings in real-world clinical settings.

Previous studies investigating MRI-based radiomics in OS have primarily focused on a single clinical outcome such as chemotherapy response or survival rate. A multicenter T1 post-contrast MRI radiomics study reported a maximum AUC of 0.88 for predicting therapy response [32]. A study using T2-weighted MRI radiomics reported lower predictive performance (AUC = 0.708) [25]. For survival outcomes, published C-indices range from 0.741 (T2-weighted MRI radiomics) [25] to 0.813 (diffusion MRI radiomics) [28]. Although direct comparison is limited due to differences in evaluation metrics, our study’s C-index values are expected to exceed this range given our excellent ROC AUC. Our study has also demonstrated comparatively higher AUC values. Moreover, previous analyses were often not pediatric-specific and relied on whole-tumor segmentation alone, overlooking the spatial heterogeneity and tissue-specific behavior of OS. Standardized feature extraction protocols, such as those recommended by the IBSI, were not consistently applied or reported in earlier studies, potentially affecting the reproducibility and comparability of radiomic studies. In contrast, our study adopts an IBSI-compliant radiomics pipeline and applies image filters to enhance feature sensitivity. In addition, we introduced multi-region segmentation strategies (including tumor sampling and bone/soft tissue separation) to better account for structural heterogeneity in pediatric OS. By jointly modeling multiple clinical outcomes and using more biologically informed segmentation, our work offers a more comprehensive and reproducible approach, thereby advancing radiomics research in OS. Notably, models based on bone and soft tissue segmentation consistently outperformed those using whole-tumor or tumor-sampling segmentation, despite the smaller sample size. Certain RFs from bone and soft tissue regions, including filtered features that capture textural patterns imperceptible to the human eye, emerged as critical predictors across multiple outcomes. These results highlight the value of capturing distinct growth patterns in bone and surrounding soft tissues that are characteristic of OS and may reflect different biological processes. Clinical features, such as presence of metastasis at diagnosis, skip lesions, progressive disease, and OS type, further enhanced model performance. In contrast, models excluding RFs often required more features and demonstrated lower predictive performance (max 45% in validation ROC and 33% in testing ROC).

Despite these promising findings, our study has limitations. First, the manual effort required for bone/soft tissue segmentation constrained the training sample size and precluded external validation of these models. Automated segmentation tools could address this limitation in future research, reducing labor demands and facilitating broader validation efforts. Second, due to resource limitations and the availability of only one radiologist, intra- or inter-observer variability testing was not performed. Third, our exclusive reliance on post-contrast T1-weighted MRI may have overlooked valuable information from other sequences, such as T2-weighted or diffusion-weighted imaging, which might provide additional insights into tumor characteristics and heterogeneity. Future studies incorporating multimodal imaging data are desired.

In the future, we aim to validate these findings using larger, multicenter datasets and incorporating additional imaging modalities and longitudinal time factors. Moreover, exploring automated feature extraction and segmentation methods will be essential for enhancing reproducibility and scalability.

## 5. Conclusions

Our findings contribute to the growing evidence supporting the utility of radiomics in pediatric OS. We have demonstrated the potential of T1w-MRI-derived radiomics and certain pre-treatment variables for improving the prediction of critical clinical outcomes. Uniquely, this study addresses multiple outcomes in a pediatric population simultaneously using an IBSI-compliant radiomics framework, including multi-region segmentation strategies to capture tumor heterogeneity more effectively. Our work underscores the need for reproducible radiomics pipelines in future studies. The approach has the potential to support pediatric OS clinical risk stratification, inform treatment planning, and ultimately enable more tailored treatment strategies to improve patient outcomes.

## Figures and Tables

**Figure 1 cancers-17-02586-f001:**
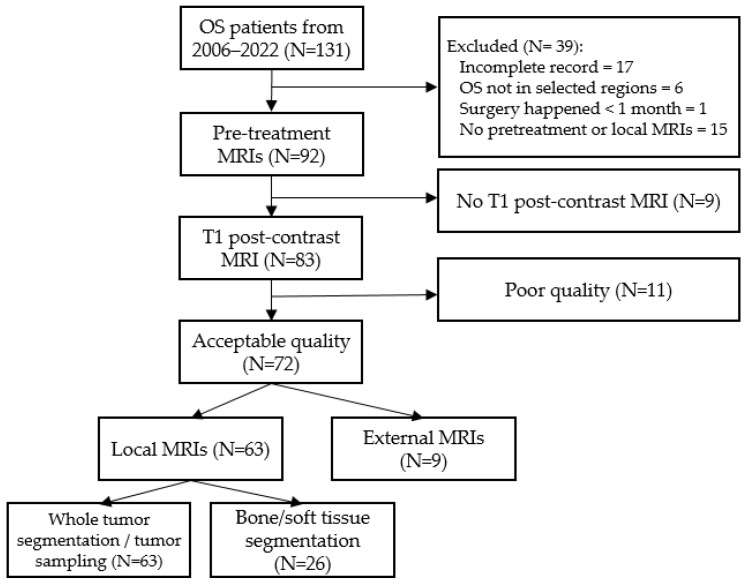
Recruitment flowchart.

**Figure 2 cancers-17-02586-f002:**
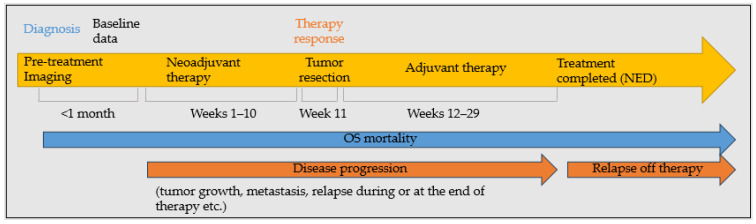
Data collection timeline.

**Figure 3 cancers-17-02586-f003:**
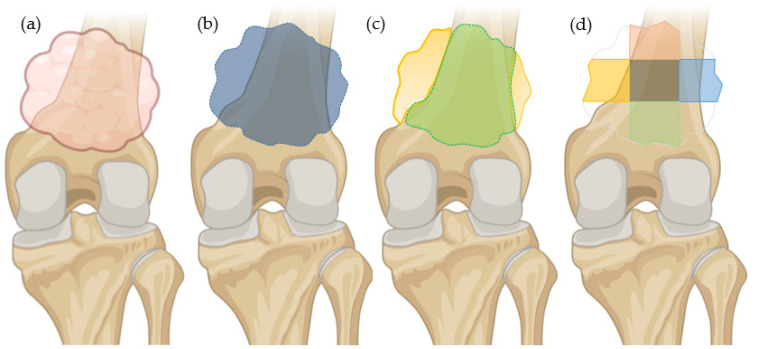
Comparison between three segmentation methods, (**a**) OS tumor without mask, (**b**) mask for whole-tumor segmentation (blue = whole tumor), (**c**) mask for bone/soft tissue segmentation (yellow = soft tissue; green = bone), (**d**) mask for tumor sampling (7 non-overlapping regions from front (not displayed), back (not displayed), top (orange), bottom (green), left (yellow), right (blue), and middle region (dark green), respectively).

**Figure 4 cancers-17-02586-f004:**
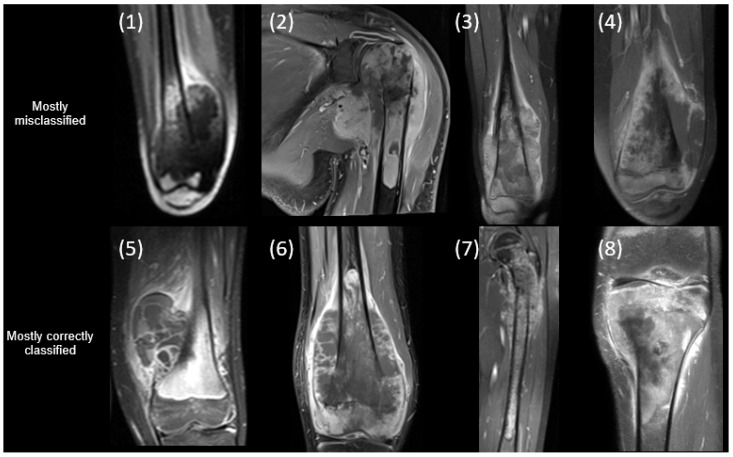
Example classification results for eight patients: four frequently misclassified and four consistently correctly classified by our models. Each panel displays the middle slice from a patient’s MRI. Patient 1 was misclassified by 26 out of 72 classifiers across different segmentation methods, feature types, and outcomes. Patient 2 was misclassified by 19 classifiers across different segmentation methods (whole-tumor segmentation and tumor sampling), feature types, and outcomes. Patient 3 was misclassified by 15 classifiers across different segmentation methods, feature types, and outcomes. Patient 4 was misclassified by 20 classifiers across different segmentation methods, feature types, and outcomes. Patient 5 was correctly classified by all classifiers except the classifier using whole-tumor segmentation with IBSI RFs only for predicting therapy response. Patient 6 was correctly classified by all classifiers except (1) the classifier using whole-tumor segmentation with IBSI RFs only for predicting progressive disease and (2) classifiers using whole-tumor segmentation with all RFs for predicting therapy response. Patient 7 was correctly classified by all classifiers except the classifier using tumor sampling, IBSI RFs, and baseline clinical features for predicting OS-related mortality. Patient 8 was correctly classified by all classifiers except (1) classifiers using whole-tumor segmentation for predicting OS-related mortality and (2) the classifier using tumor sampling with IBSI RFs only for predicting relapse off therapy.

**Table 1 cancers-17-02586-t001:** Summary statistics.

	63 Patients	26 Patients	9 Patients	*p*-Value
Age (Median/Mean ± SD)	12.29/11.82 ± 3.53	12.21/11.83 ± 3.98	11.49/12.73 ± 4.20	0.9263
Sex (Male)	43 (68.25%)	18 (69.23%)	7 (77.78%)	0.8451
Race				0.5845
Caucasian	52 (82.54%)	22 (84.62%)	6 (66.66%)	
Black/African American	9 (14.29%)	3 (11.54%)	3 (33.33%)	
Others	2 (3.17%)	1 (3.85%)	0 (0%)	
Hispanics	31 (49.21%)	14 (53.85%)	4 (44.44%)	0.8690
Laterality (right)	36 (57.14%)	16 (61.54%)	1 (11.11%)	0.0234
OS location				0.0532
Femur	41 (65.08%)	23 (88.46%)	5 (55.55%)	
Tibia	14 (22.22%)	3 (11.54%)	1 (11.11%)	
Humus	6 (9.52%)	0 (0%)	1 (11.11%)	
Fibula	2 (3.17%)	0 (0%)	2 (22.22%)	
Histological subtype				
Osteoblastic	50 (79.37%)	21 (80.77%)	9 (100%)	0.4017
Chondroblastic	21 (33.33%)	10 (38.46%)	2 (22.22%)	0.6798
Telangiectatic	5 (7.94%)	1 (3.85%)	1 (11.11%)	0.5812
Metastasis (Yes)	13 (20.63%)	7 (26.92%)	5 (55.55%)	0.0784
Skip lesion (Yes)	8 (12.70%)	4 (15.38%)	1 (11.11%)	0.8969
Progressive disease during therapy (Yes)	17 (26.98%)	6 (23.08%)	4 (44.44%)	0.4589
% necrosis (Median/Mean ± SD)	93/77.32 ± 27.08	95/81.21 ± 24.98	95/81.11 ± 26.30	0.7054
Response on therapy (Adequate)	37 (58.73%)	16 (61.54%)	6 (66.66%)	0.9509
Relapse off therapy (Yes)	16 (25.40%)	7 (26.92%)	4 (44.44%)	0.4871
OS related mortality (Yes)	19 (30.16%)	6 (23.08%)	5 (55.55%)	0.1885

Legend: Note that histological subtypes were not mutually exclusive. *p*-values were calculated for each subtype.

**Table 2 cancers-17-02586-t002:** Associations between outcome and major clinical features.

	Adequate Response to Therapy (*n* = 37)	Poor Response to Therapy (*n* = 26)	*p*-Value
Progressive disease during therapy (Yes)	3 (8.11%)	14 (53.85%)	<0.0001
Metastasis at diagnosis (Yes)	11 (29.73%)	2 (7.69%)	0.0557
Skip lesion (Yes)	7 (18.92%)	1 (3.85%)	0.1254
Osteoblastic (Yes)	30 (81.08%)	20 (76.92%)	0.6881
Chondroblastic (Yes)	9 (24.32%)	12 (46.15%)	0.0704
	Relapse off therapy (*n* = 16)	No relapse off therapy (*n* = 47)	*p*-value
Progressive disease during therapy (Yes)	4 (25%)	13 (27.66%)	0.9999
Response to therapy (Adequate)	12 (75%)	25 (53.19%)	0.1519
% necrosis (mean ± SD)	81.13 ± 28.78	76.03 ± 26.97	0.3378
Metastasis at diagnosis (Yes)	7 (43.75%)	6 (12.77%)	0.0082
Skip lesion (Yes)	4 (25%)	4 (8.51%)	0.1857
Osteoblastic (Yes)	13 (81.25%)	37 (78.72%)	0.9999
Chondroblastic (Yes)	7 (43.75%)	14 (29.79%)	0.3062
	Deceased (*n* = 19)	Alive (*n* = 44)	*p*-value
Progressive disease during therapy (Yes)	13 (68.42%)	4 (9.09%)	<0.0001
Response to therapy (Adequate)	8 (42.11%)	29 (65.91%)	0.0782
Relapse off therapy (Yes)	9 (47.37%)	7 (15.91%)	0.0085
% necrosis (mean ± SD)	66.58 ± 28.48	81.96 ± 25.72	0.0762
Metastasis at diagnosis (Yes)	7 (36.84%)	6 (13.64%)	0.0367
Skip lesion (Yes)	4 (21.05%)	4 (9.09%)	0.2286
Osteoblastic (Yes)	16 (84.21%)	34 (77.27%)	0.7375
Chondroblastic (Yes)	10 (52.63%)	11 (25%)	0.0327

**Table 3 cancers-17-02586-t003:** Classification performance for progressive disease.

Segmentation	RF Type	Validation Set					Testing Set					Best Classifier	No. Features	External Set		
		Accuracy	Sensitivity	Segmentation	ROC AUC	PR AUC	Accuracy	Sensitivity	Specificity	ROC AUC	PR AUC			Accuracy	Sensitivity	Specificity
Whole tumor	IBSI RFs	0.84 ± 0.14	0.70 ± 0.27	0.89 ± 0.13	0.68 ± 0.29	0.66 ± 0.29	0.69	0.50	0.78	0.67	0.69	MLP	4	0.33	0.25	0.40
	IBSI RFs + baseline clinical	0.90 ± 0.11	0.93 ± 0.13	0.89 ± 0.11	0.81 ± 0.17	0.87 ± 0.16	0.69	0.25	0.89	0.58	0.41	SVM rbf	8	0.44	0.25	0.60
	All RFs	0.92 ± 0.08	0.93 ± 0.13	0.91 ± 0.07	0.88 ± 0.13	0.91 ± 0.10	0.77	0.50	0.89	0.67	0.65	KNN (k = 3)	13	0.44	0.25	0.60
	All RFs + baseline clinical	0.96 ± 0.05	0.93 ± 0.13	0.97 ± 0.06	0.92 ± 0.09	0.94 ± 0.08	0.77	0.50	0.89	0.51	0.56	Random forest	14	0.56	0.25	0.80
Whole tumor/tumor sampling	Only baseline clinical	0.78 ± 0.20	0.93 ± 0.13	0.71 ± 0.30	0.70 ± 0.26	0.71 ± 0.19	0.77	0.75	0.78	0.88	0.78	Random forest	12	0.22	0.50	0.00
Tumor sampling	IBSI RFs	0.82 ± 0.15	0.87 ± 0.16	0.81 ± 0.18	0.75 ± 0.23	0.80 ± 0.20	0.85	0.75	0.89	0.69	0.68	MLP	15	0.44	0.25	0.60
	IBSI RFs + baseline clinical	0.90 ± 0.09	1.0 ± 0	0.87 ± 0.13	0.84 ± 0.17	0.93 ± 0.09	0.69	0.50	0.78	0.63	0.56	Gradient boosting	6	0.67	0.50	0.80
	All RFs	0.92 ± 0.12	1.0 ± 0	0.89 ± 0.17	0.91 ± 0.12	0.94 ± 0.08	0.31	1.00	0.00	0.86	0.81	MLP	15	0.44	1.00	0.00
	All RFs + baseline clinical	0.92 ± 0.12	0.93 ± 0.13	0.93 ± 0.15	0.90 ± 0.13	0.92 ± 0.09	0.31	1.00	0.00	0.89	0.83	MLP	11	0.44	1.00	0.00
Bone/soft tissue	IBSI RFs	0.95 ± 0.07	1.0 ± 0	0.93 ± 0.09	0.94 ± 0.08	0.97 ± 0.05	0.83	0.50	1.00	0.88	0.83	LDA	1	-	-	-
	IBSI RFs + baseline clinical	0.95 ± 0.07	1.0 ± 0	0.93 ± 0.09	0.94 ± 0.08	0.97 ± 0.05	0.83	0.50	1.00	0.88	0.83	LDA	1	-	-	-
	All RFs	1.0 ± 0	1.0 ± 0	1.0 ± 0	1.0 ± 0	1.0 ± 0	0.67	0.50	0.75	0.75	0.75	SVM rbf	14	-	-	-
	All RFs + baseline clinical	1.0 ± 0	1.0 ± 0	1.0 ± 0	1.0 ± 0	1.0 ± 0	0.67	0.50	0.75	0.81	0.58	Random forest	15	-	-	-
	Only baseline clinical	0.79 ± 0.21	1.0 ± 0	0.73 ± 0.25	0.69 ± 0.32	0.82 ± 0.23	0.67	0.00	1.00	0.75	0.75	MLP	9	-	-	-

**Table 4 cancers-17-02586-t004:** Classification performance for therapy response.

Segmentation	RF Type	Validation Set					Testing Set					Best Classifier	No. Features	External Set		
		Accuracy	Sensitivity	Specificity	ROC AUC	PR AUC	Accuracy	Sensitivity	Specificity	ROC AUC	PR AUC			Accuracy	Sensitivity	Specificity
Whole tumor	IBSI RFs	0.72 ± 0.04	0.59 ± 0.12	0.91 ± 0.11	0.83 ± 0.04	0.73 ± 0.13	0.54	0.25	1.00	0.69	0.79	KNN (k = 10)	3	0.44	0.50	0.33
	IBSI RFs + baseline clinical	0.78 ± 0.13	0.67 ± 0.24	0.95 ± 0.10	0.86 ± 0.09	0.73 ± 0.17	0.69	0.63	0.80	0.80	0.87	MLP	9	0.56	0.50	0.67
	All RFs	0.88 ± 0.08	0.79 ± 0.14	1.0 ± 0	0.93 ± 0.06	0.87 ± 0.12	0.62	0.63	0.60	0.78	0.88	SVM rbf	14	0.56	0.67	0.33
	All RFs + baseline clinical	0.88 ± 0.08	0.79 ± 0.14	1.0 ± 0	0.93 ± 0.06	0.87 ± 0.12	0.62	0.63	0.60	0.78	0.88	SVM rbf	14	0.56	0.67	0.33
Whole tumor/tumor sampling	Only baseline clinical	0.76 ± 0.08	0.76 ± 0.17	0.76 ± 0.16	0.84 ± 0.06	0.75 ± 0.10	0.77	0.88	0.60	0.90	0.95	SVM sigmoid	15	0.67	1.00	0.00
Tumor sampling	IBSI RFs	0.80 ± 0.06	0.69 ± 0.12	0.96 ± 0.08	0.89 ± 0.06	0.84 ± 0.04	0.62	0.75	0.40	0.68	0.77	MLP	5	0.67	0.83	0.33
	IBSI RFs + baseline clinical	0.88 ± 0.08	0.83 ± 0.11	0.95 ± 0.10	0.94 ± 0.05	0.89 ± 0.07	0.62	0.50	0.80	0.76	0.84	Random forest	5	0.44	0.50	0.33
	All RFs	0.94 ± 0.05	0.93 ± 0.09	0.95 ± 0.10	0.98 ± 0.02	0.97 ± 0.03	0.46	0.25	0.80	0.60	0.72	SVM Polynomial	11	0.33	0.00	1.00
	All RFs + baseline clinical	0.94 ± 0.05	0.93 ± 0.09	0.95 ± 0.10	0.98 ± 0.02	0.97 ± 0.03	0.46	0.25	0.80	0.60	0.72	SVM Polynomial	11	0.33	0.00	1.00
Bone/soft tissue	IBSI RFs	1.0 ± 0	1.0 ± 0	1.0 ± 0	1.0 ± 0	1.0 ± 0	0.50	0.25	1.00	0.75	0.92	Random forest	10	-	-	-
	IBSI RFs + baseline clinical	1.0 ± 0	1.0 ± 0	1.0 ± 0	1.0 ± 0	1.0 ± 0	0.33	0.25	0.50	0.63	0.80	Random forest	12	-	-	-
	All RFs	1.0 ± 0	1.0 ± 0	1.0 ± 0	1.0 ± 0	1.0 ± 0	1.00	1.00	1.00	1.00	1.00	KNN (k = 5)	4	-	-	-
	All RFs + baseline clinical	1.0 ± 0	1.0 ± 0	1.0 ± 0	1.0 ± 0	1.0 ± 0	0.83	1.00	0.50	0.88	0.95	LDA	3	-	-	-
	Only baseline clinical	0.90 ± 0.12	0.83 ± 0.21	1.0 ± 0	0.93 ± 0.10	0.83 ± 0.21	0.50	0.25	1.00	0.75	0.92	Logistic regression	11	-	-	-

**Table 5 cancers-17-02586-t005:** Classification performance for relapse off therapy.

Segmentation	RF Type	Validation Set					Testing Set					Best Classifier	No. Features	External Set		
		Accuracy	Sensitivity	Specificity	ROC AUC	PR AUC	Accuracy	Sensitivity	Specificity	ROC AUC	PR AUC			Accuracy	Sensitivity	Specificity
Whole tumor	IBSI RFs	0.84 ± 0.10	0.77 ± 0.20	0.86 ± 0.16	0.70 ± 0.17	0.73 ± 0.18	0.77	0.00	1.00	0.73	0.42	SVM sigmoid	4	0.67	0.25	1.00
	IBSI RFs + baseline clinical	0.80 ± 0.09	0.80 ± 0.27	0.81 ± 0.18	0.67 ± 0.11	0.80 ± 0.09	0.69	0.33	0.80	0.58	0.35	Random forest	5	0.44	0.50	0.40
	IBSI RFs + baseline clinical + prior outcomes	0.82 ± 0.08	0.70 ± 0.27	0.87 ± 0.16	0.65 ± 0.14	0.75 ± 0.12	0.77	0.00	1.00	0.63	0.39	Logistic regression	3	0.56	0.00	1.00
	All RFs	0.84 ± 0.10	0.93 ± 0.13	0.81 ± 0.17	0.75 ± 0.18	0.84 ± 0.11	0.54	0.33	0.60	0.33	0.22	SVM rbf	11	0.78	0.75	0.80
	All RFs + baseline clinical	0.88 ± 0.10	1.0 ± 0	0.83 ± 0.14	0.89 ± 0.11	0.92 ± 0.08	0.46	0.33	0.50	0.57	0.36	SVM rbf	10	0.67	1.00	0.40
	All RFs + baseline clinical + prior outcomes	0.90 ± 0.09	0.87 ± 0.27	0.92 ± 0.10	0.78 ± 0.25	0.84 ± 0.20	0.62	1.00	0.50	0.73	0.41	MLP	5	0.56	1.00	0.20
Whole tumor/tumor sampling	Only baseline clinical	0.74 ± 0.19	0.93 ± 0.13	0.68 ± 0.24	0.68 ± 0.26	0.75 ± 0.20	0.62	0.67	0.60	0.58	0.53	Random forest	9	0.56	0.25	0.80
	Only baseline clinical + prior outcomes	0.78 ± 0.22	0.93 ± 0.13	0.74 ± 0.29	0.72 ± 0.29	0.76 ± 0.28	0.77	0.00	1.00	0.57	0.30	MLP	8	0.56	0.00	1.00
Tumor sampling	IBSI RFs	0.84 ± 0.16	0.87 ± 0.16	0.83 ± 0.23	0.80 ± 0.18	0.78 ± 0.18	0.77	0.33	0.90	0.67	0.43	SVM sigmoid	11	0.44	0.00	0.80
	IBSI RFs + baseline clinical	0.94 ± 0.05	0.93 ± 0.13	0.95 ± 0.07	0.86 ± 0.13	0.93 ± 0.06	0.77	0.33	0.90	0.73	0.61	SVM polynomial	13	0.56	0.25	0.80
	IBSI RFs + baseline clinical + prior outcomes	0.94 ± 0.05	0.93 ± 0.13	0.95 ± 0.07	0.86 ± 0.13	0.93 ± 0.06	0.77	0.33	0.90	0.73	0.61	SVM polynomial	13	0.56	0.25	0.80
	All RFs	0.92 ± 0.08	0.93 ± 0.13	0.92 ± 0.11	0.90 ± 0.08	0.93 ± 0.07	0.23	1.00	0.00	0.73	0.41	MLP	15	0.44	1.00	0.00
	All RFs + baseline clinical	0.94 ± 0.05	1.0 ± 0	0.92 ± 0.07	0.93 ± 0.07	0.97 ± 0.03	0.31	1.00	0.10	0.67	0.64	MLP	10	0.44	1.00	0.00
	All RFs + baseline clinical + prior outcomes	0.94 ± 0.05	1.0 ± 0	0.92 ± 0.07	0.93 ± 0.07	0.97 ± 0.03	0.31	1.00	0.10	0.67	0.64	MLP	10	0.44	1.00	0.00
Bone/soft tissue	IBSI RFs	0.95 ± 0.07	1.0 ± 0	0.93 ± 0.09	0.94 ± 0.08	0.97 ± 0.05	0.67	0.50	0.75	0.75	0.75	Logistic regression	3	-	-	-
	IBSI RFs + baseline clinical	1.0 ± 0	1.0 ± 0	1.0 ± 0	1.0 ± 0	1.0 ± 0	0.67	0.50	0.75	0.75	0.75	SVM rbf	3	-	-	-
	IBSI RFs + baseline clinical + prior outcomes	1.0 ± 0	1.0 ± 0	1.0 ± 0	1.0 ± 0	1.0 ± 0	0.67	0.50	0.75	0.75	0.75	SVM rbf	3	-	-	-
	All RFs	1.0 ± 0	1.0 ± 0	1.0 ± 0	1.0 ± 0	1.0 ± 0	0.50	0.50	0.50	0.63	0.50	MLP	15	-	-	-
	All RFs + baseline clinical	1.0 ± 0	1.0 ± 0	1.0 ± 0	1.0 ± 0	1.0 ± 0	0.50	0.50	0.50	0.75	0.75	SVM rbf	10	-	-	-
	All RFs + baseline clinical + prior outcomes	1.0 ± 0	1.0 ± 0	1.0 ± 0	1.0 ± 0	1.0 ± 0	0.50	0.50	0.50	0.75	0.75	SVM rbf	10	-	-	-
	Only baseline clinical	0.95 ± 0.07	1.0 ± 0	0.93 ± 0.09	0.94 ± 0.08	0.97 ± 0.05	0.67	0.50	0.75	0.75	0.58	MLP	11	-	-	-
	Only baseline clinical + prior outcomes	0.91 ± 0.14	1.0 ± 0	0.87 ± 0.19	0.92 ± 0.12	0.93 ± 0.09	0.67	0.50	0.75	0.75	0.75	SVM sigmoid	2	-	-	-

**Table 6 cancers-17-02586-t006:** Classification performance for OS-related mortality.

Segmentation	RF Type	Validation Set					Testing Set					Best Classifier	No. Features	External Set		
		Accuracy	Sensitivity	Specificity	ROC AUC	PR AUC	Accuracy	Sensitivity	Specificity	ROC AUC	PR AUC			Accuracy	Sensitivity	Specificity
Whole tumor	IBSI RFs	0.80 ± 0.13	0.80 ± 0.16	0.80 ± 0.25	0.69 ± 0.16	0.73 ± 0.09	0.31	0.75	0.11	0.47	0.35	SVM sigmoid	10	0.67	1.00	0.25
	IBSI RFs + baseline clinical	0.88 ± 0.08	0.73 ± 0.25	0.94 ± 0.11	0.73 ± 0.18	0.77 ± 0.17	0.69	0.25	0.89	0.31	0.33	KNN (k = 15)	8	0.56	0.20	1.00
	IBSI RFs + baseline clinical + prior outcomes	0.98 ± 0.04	1.0 ± 0	0.97 ± 0.06	0.98 ± 0.03	0.99 ± 0.02	0.85	1.00	0.78	0.92	0.85	SVM rbf	3	0.56	0.40	0.75
	All RFs	0.92 ± 0.12	0.93 ± 0.13	0.91 ± 0.11	0.91 ± 0.15	0.92 ± 0.13	0.85	0.75	0.89	0.86	0.75	Random forest	15	0.89	1.00	0.75
	All RFs + baseline clinical	0.92 ± 0.04	0.80 ± 0.16	0.97 ± 0.06	0.87 ± 0.08	0.87 ± 0.11	0.77	0.75	0.78	0.81	0.57	Random forest	6	0.56	0.20	1.00
	All RFs + baseline clinical + prior outcomes	1.0 ± 0	1.0 ± 0	1.0 ± 0	1.0 ± 0	1.0 ± 0	0.92	1.00	0.89	0.97	0.95	KNN (k = 5)	6	1.00	1.00	1.00
whole tumor/tumor sampling	Only baseline clinical	0.82 ± 0.12	0.73 ± 0.13	0.86 ± 0.16	0.69 ± 0.21	0.69 ± 0.19	0.62	0.50	0.67	0.67	0.46	SVM polynomial	15	0.44	0.20	0.75
	Only baseline clinical + prior outcomes	0.98 ± 0.04	1.0 ± 0	0.97 ± 0.06	0.98 ± 0.03	0.99 ± 0.02	0.85	1.00	0.78	0.92	0.85	SVM rbf	3	0.56	0.40	0.75
Tumor sampling	IBSI RFs	0.90 ± 0.06	0.73 ± 0.25	0.97 ± 0.06	0.80 ± 0.15	0.81 ± 0.18	0.62	0.25	0.78	0.61	0.53	Random forest	5	0.33	0.00	0.75
	IBSI RFs + baseline clinical	0.84 ± 0.10	0.87 ± 0.16	0.83 ± 0.17	0.77 ± 0.17	0.84 ± 0.14	0.69	0.50	0.78	0.72	0.47	Random forest	7	0.33	0.00	0.75
	IBSI RFs + baseline clinical + prior outcomes	0.98 ± 0.04	1.0 ± 0	0.97 ± 0.06	0.98 ± 0.03	0.99 ± 0.02	0.85	1.00	0.78	0.92	0.85	SVM rbf	3	0.56	0.40	0.75
	All RFs	0.94 ± 0.08	1.0 ± 0	0.91 ± 0.11	0.91 ± 0.11	0.95 ± 0.06	0.39	1.00	0.11	0.81	0.67	Random forest	12	0.67	1.00	0.25
	All RFs + baseline clinical	0.94 ± 0.08	0.93 ± 0.13	0.94 ± 0.11	0.91 ± 0.11	0.93 ± 0.08	0.62	1.00	0.44	0.85	0.73	Random forest	5	0.56	1.00	0.00
	All RFs + baseline clinical + prior outcomes	1.0 ± 0	1.0 ± 0	0.94 ± 0.11	0.97 ± 0.05	0.98 ± 0.04	0.85	0.75	0.89	0.94	0.92	KNN (k = 8)	8	0.78	0.60	1.00
Bone/soft tissue	IBSI RFs	1.0 ± 0	1.0 ± 0	1.0 ± 0	1.0 ± 0	1.0 ± 0	1.00	1.00	1.00	1.00	1.00	SVM sigmoid	7	-	-	-
	IBSI RFs + baseline clinical	1.0 ± 0	1.0 ± 0	1.0 ± 0	1.0 ± 0	1.0 ± 0	1.00	1.00	1.00	1.00	1.00	KNN (k = 6)	8	-	-	-
	IBSI RFs + baseline clinical + prior outcomes	1.0 ± 0	1.0 ± 0	1.0 ± 0	1.0 ± 0	1.0 ± 0	1.00	1.00	1.00	1.00	1.00	Random forest	8	-	-	-
	All RFs	1.0 ± 0	1.0 ± 0	1.0 ± 0	1.0 ± 0	1.0 ± 0	0.83	1.00	0.75	0.88	0.83	SVM rbf	6	-	-	-
	All RFs + baseline clinical	1.0 ± 0	1.0 ± 0	1.0 ± 0	1.0 ± 0	1.0 ± 0	0.83	1.00	0.75	0.88	0.83	SVM rbf	6	-	-	-
	All RFs + baseline clinical + prior outcomes	1.0 ± 0	1.0 ± 0	1.0 ± 0	1.0 ± 0	1.0 ± 0	1.00	1.00	1.00	1.00	1.00	Random forest	4	-	-	-
	Only baseline clinical	0.73 ± 0.29	1.0 ± 0	0.68 ± 0.35	0.57 ± 0.33	0.68 ± 0.35	0.67	0.50	0.75	0.63	0.50	MLP	15	-	-	-
	Only baseline clinical + prior outcomes	1.0 ± 0	1.0 ± 0	1.0 ± 0	1.0 ± 0	1.0 ± 0	0.83	0.50	1.00	0.88	0.83	SVM rbf	4	-	-	-

**Table 7 cancers-17-02586-t007:** Demographics and clinical information of the selected patients in Figure 4.

Patient	Gender	Age (y)	Skip Lesion	Meta-Stasis	OS Type	Progressive Disease	% Necrosis	Relapse off Therapy	Mortality
1	Female	14.0	No	Yes	Osteobl.	No	100	Yes	Yes
2	Male	16.3	No	Yes	Osteobl.	No	99	No	No
3	Female	14.9	Yes	Yes	Osteobl.	Yes	>99	Yes	Yes
4	Male	10.9	No	Yes	Chondrobl.	No	>99	No	No
5	Female	9.8	No	No	Osteobl. and telangiectatic	No	100	No	No
6	Male	12.3	Yes	Yes	Osteobl. and chondrob.	Yes	95	Yes	Yes
7	Male	9.0	No	No	Osteobl.	No	87	No	No
8	Male	14.2	No	No	Osteobl.	Yes	40	No	yes

## Data Availability

The data presented in this study are available upon request from the corresponding author due to privacy and confidentiality restrictions associated with hospital records.

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
