# Peer review of "MRI-Based Radiomics for Outcome Stratification in Pediatric Osteosarcoma†"

_cancers, 2025, doi:10.3390/cancers17152586_

Round 1

Reviewer 1 Report

Comments and Suggestions for Authors

The retrospective study reports the use of filtered and Image Biomarker Standardization Initiative (IBSI) to validate radiomics using T1-weighted MRI data from 63 pediatric OS patients with extremity tumors. The study aims to enhance diagnostic, prognostic, and predictive accuracy and to identify reliable imaging biomarkers for early risk stratification and more effective treatment for pediatric OS by using radiomics data combined with advanced bioinformatics tools and a hierarchical model. Results show the new models demonstrated robust predictive values for multiple clinical outcomes, including progressive disease, therapy response, relapse occurrence, and mortality. The integration of multiple clinical features and a hierarchical approach provides a more comprehensive view of disease trajectories. Compared to most prior studies, the current strategy does not rely on whole-tumor segmentation. Instead, it considers regional variations within the tumor, which is critical due to the tumor's inherent heterogeneity. The study makes an important contribution to the field by simultaneously addressing multiple clinical outcomes and utilizing standardized radiomics features in a pediatric population. The authors need to address the following issues:

  1. The dataset for 63 patients is split into 80% training and 20% testing. How do the authors assign the training and testing groups to reduce any bias?
  2. Considering multiple parameters, what is the best-performing model to inform a reliable and reproducible clinical decision in OS patients regarding tumor progression prediction, treatment response, relapse occurrence, and patient mortality?
  3. Table 6 is missing from the manuscript. Please provide Table 6.
  4. The legend in Figure 4 needs to be reorganized, as the partial legend is included in Figure 4, while the others are presented in the context.
  5. Please provide the significance level for the statistical analysis.
  6. Please provide the full names of TCH and IRB during their first appearance. Please check the reference format to ensure consistency.
Comments on the Quality of English Language

English is acceptable. 

Author Response

Dear Reviewer 1

We would like to thank you for all the comments and time spent on our manuscript. We have prepared a revised manuscript addressing the comments and provided itemized answers in this document. We believe that we have answered all the questions and clarified the manuscript text. Please find below itemized responses to yours comments written in a blue font.

Comments and Suggestions for Authors

The retrospective study reports the use of filtered and Image Biomarker Standardization Initiative (IBSI) to validate radiomics using T1-weighted MRI data from 63 pediatric OS patients with extremity tumors. The study aims to enhance diagnostic, prognostic, and predictive accuracy and to identify reliable imaging biomarkers for early risk stratification and more effective treatment for pediatric OS by using radiomics data combined with advanced bioinformatics tools and a hierarchical model. Results show the new models demonstrated robust predictive values for multiple clinical outcomes, including progressive disease, therapy response, relapse occurrence, and mortality. The integration of multiple clinical features and a hierarchical approach provides a more comprehensive view of disease trajectories. Compared to most prior studies, the current strategy does not rely on whole-tumor segmentation. Instead, it considers regional variations within the tumor, which is critical due to the tumor's inherent heterogeneity. The study makes an important contribution to the field by simultaneously addressing multiple clinical outcomes and utilizing standardized radiomics features in a pediatric population. The authors need to address the following issues:

Comments 1: The dataset for 63 patients is split into 80% training and 20% testing. How do the authors assign the training and testing groups to reduce any bias?

Response 1: Thank you for this question. To mitigate selection bias and ensure reproducibility, we performed the data split using the scikit-learn Python package, specifically the train_test_split function, with a fixed random seed. This approach ensures that the split is random but reproducible across runs. The test set was defined once and remained completely untouched throughout the model development process. To reduce potential overfitting and enhance model robustness, we applied 5-fold cross-validation (or 3-fold for bone/tissue segmentation) within the training set for model selection and hyperparameter tuning. After we selected the ‘optimal’ model from the training data, we evaluated its performance on the held-out test set.

To further address potential bias from class imbalance, we ensured that the proportions of outcome classes (e.g., mortality, relapse) were preserved in both training and testing sets by using stratified sampling based on the outcome variable. This approach helps maintain a representative distribution of outcomes across the splits, especially important given the modest sample size.

Comments 2: Considering multiple parameters, what is the best-performing model to inform a reliable and reproducible clinical decision in OS patients regarding tumor progression prediction, treatment response, relapse occurrence, and patient mortality?

Response 2: Thank you for this insightful question. A key characteristic of our study is that we approached each clinical outcome independently and within a hierarchical clinical framework. The latter design reflects the real-world progression of disease and care decisions in pediatric osteosarcoma.

Because of this staged and outcome-specific modeling strategy, there is not a single best model architecture or feature set that generalizes across all endpoints. Each outcome required its own tailored combination of features and classifier, selected through cross-validated training and tested on a held-out set.

However, here is the list of the best classifier(s) based on the validation ROC AUC for each outcome presented in a table below:

Outcome

Segmentation

Feature types

# features

Best classifier

Validation AUC ROC

Testing AUC ROC

Disease progression

Bone/ soft tissue

All RFs + baseline clinical

15

Random forest

1.0±0

0.81

Treatment response

Bone/ soft tissue

All RFs only

4

KNN (k=5)

1.0±0

1.0

Relapse occurrence

Bone/ soft tissue

IBSI RFs + baseline clinical

3

SVM rbf

1.0±0

0.75

OS Mortality

Bone/ soft tissue

IBSI RFs only

7

SVM sigmoid

1.0±0

1.0

Comments 3: Table 6 is missing from the manuscript. Please provide Table 6.

Response 3: Thank you for pointing that out. Table 6 has been added to the revised manuscript.

Comments 4: The legend in Figure 4 needs to be reorganized, as the partial legend is included in Figure 4, while the others are presented in the context.

Response 4: Thank you for pointing that out. The legend of figure 4 has been revised in the latest manuscript.

Comments 5: Please provide the significance level for the statistical analysis.

Response 5: Thank you for pointing out the missing significance level. The significance level for all statistical analyses is set to be 0.05, and it’s added to the method Statistical analysis section.

Comments 6: Please provide the full names of TCH and IRB during their first appearance. Please check the reference format to ensure consistency.

Response 6: Thank you for the suggestion. We have added the full name of the Institutional Review Board (IRB) to its first mention in the manuscript. To maintain institutional anonymity, we have replaced “TCH” in Figure 1 with “local”.

Reviewer 2 Report

Comments and Suggestions for Authors

This paper investigates how MRI-based radiomics features, combined with machine learning, can predict clinical outcomes in pediatric osteosarcoma patients, including progression, therapy response, relapse, and mortality. The authors describe an interesting and helpful line of research.

Please see some of the major concerns listed below:

Materials and Methods:

Could the authors clarify the years of experience of the radiologist who performed the segmentations, and whether reproducibility (intra- or interobserver variability) was tested?

Where exactly were the computational analyses run (hardware specifications, GPU vs CPU), and how long did training and testing typically take? This is important for future reproducibility.

Was the external dataset used only for testing, or also for hyperparameter tuning at any point?

Given the relatively small sample size, did the authors perform any data augmentation or bootstrapping to stabilize training, or apply techniques to mitigate overfitting?

How did the authors decide on the cut-offs for feature correlation (0.8 or 0.9) in the reduction process, and could different thresholds alter feature selection significantly?

Discussion:

Could the authors discuss how these models might perform with multimodal imaging (adding T2-weighted or diffusion sequences), which is increasingly relevant in osteosarcoma staging?

It would also strengthen the paper to compare directly to previous pediatric OS radiomics studies on therapy response or survival and more explicitly state what is novel beyond adding multi-region segmentation.

Author Response

Dear Reviewer 2,

We would like to thank you for all the comments and time spent on our manuscript. We have prepared a revised manuscript addressing the comments and provided itemized answers in this document. We believe that we have answered all the questions and clarified the manuscript text. Please find below itemized responses to yours comments written in a blue font.

Comments and Suggestions for Authors

This paper investigates how MRI-based radiomics features, combined with machine learning, can predict clinical outcomes in pediatric osteosarcoma patients, including progression, therapy response, relapse, and mortality. The authors describe an interesting and helpful line of research. Please see some of the major concerns listed below:

Materials and Methods:

Comments 1: Could the authors clarify the years of experience of the radiologist who performed the segmentations, and whether reproducibility (intra- or interobserver variability) was tested?

Response 1: Thank you for this important question. The bone/ soft tissue segmentations were performed by a single pediatric radiologist with 9 years of experience. The whole tumor segmentation was subsequently conducted by a postdoctoral researcher with 2 years of experience in medical image analysis, building upon the initial bone/tissue segmentations. We have added details of their experience to the manuscript’s Segmentation Method section.

Due to resource limitations and the availability of only one radiologist, formal intra- or interobserver variability testing was not performed. However, all segmentations followed standardized protocols to maintain consistency across the dataset, especially the tumor sampling method. A limitation of having a single radiologist perform the segmentations has been acknowledged in the manuscript.

Comments 2: Where exactly were the computational analyses run (hardware specifications, GPU vs CPU), and how long did training and testing typically take? This is important for future reproducibility.

Response 2: Thank you for asking about the computational aspects of the analysis. The computational analyses were performed on a Linux workstation equipped with an Intel(R) Core (TM) i9-9900K CPU @ 3.60 GHz (16 logical cores), 64 GB of RAM, and a single NVIDIA GeForce RTX 1080 Ti GPU with 11GiB memory. The system typically had ~ 50 GB of available memory during analysis. Training times varied depending on the feature set and segmentation approach used. Typically, an outcome- and segmentation-specific analysis required 1.5 to 4.5 hours to complete. For example, the shortest runtimes were observed when using IBRI radiomic features combined with whole tumor segmentation, while analyses including all radiomic features and tumor sampling segmentation required the longest processing times. Information on computational resources has been included in the Method Machine learning section, and analysis times are included in the Results section.

Comments 3: Was the external dataset used only for testing, or also for hyperparameter tuning at any point?

Response 3: Thank you for raising this important point regarding the use of external data. Given the limited size of the external dataset, it was used exclusively for testing purposes. The external set served as a true “unseen” dataset, providing a realistic estimate of performance on new patients. All model training and hyperparameter tuning were conducted solely on the internal training data. Nevertheless, the image processing and standardization pipelines applied to the external dataset were consistent with those used for the internal data.

Comments 4: Given the relatively small sample size, did the authors perform any data augmentation or bootstrapping to stabilize training, or apply techniques to mitigate overfitting?

Response 4: Thank you for this question. To mitigate overfitting and obtain robust performance estimates from our relatively small dataset, we applied 5-fold cross-validation (or 3-fold in extreme cases) during model training. In addition, we limited the number of features used per classifier (max 15). We applied feature selection techniques to identify and retain only the most informative radiomic features for each outcome. This approach further reduces model complexity and helps prevent overfitting by eliminating irrelevant or redundant variables that could introduce noise. The threshold of 15 was selected based on our experience that model performance showed minimal improvement beyond this number, while also considering the balance between the number of features and the sample size. One of the classifiers examined, Random Forest, is an ensemble method inherently designed to reduce overfitting.

Given class imbalance in some outcomes, we initially explored the Synthetic Minority Over-sampling Technique (SMOTE). However, this approach led to poorer model performance, likely due to artifacts introduced by synthetic samples. Thus, we maintained consistent outcome proportions between training and testing sets using stratified splitting and reported performance metrics such as the area under the ROC curve (AUC) rather than accuracy to better reflect model discrimination.

While image-level data augmentation can enhance sample size and model robustness, it was not applied here as radiomic analysis can work on a much smaller sample size than deep learning. Here are some recent radiomics studies that have been conducted with sample sizes smaller than ours, including the following:   Lauer D. 2024, DOI:  10.1172/jci.insight.181757, Ahrari S. 2024, DOI: 10.1038/s41598-024-53693-x, and Esposito F. 2025, DOI: 10.3390/diagnostics15101281.

Comments 5: How did the authors decide on the cut-offs for feature correlation (0.8 or 0.9) in the reduction process, and could different thresholds alter feature selection significantly?

Response 5: Thank you for this question. The goal of applying feature reduction via correlation analysis was to minimize redundancy and improve the stability of subsequent model training steps. The total number of features varied by segmentation approach. For IBSI RFs, with region sampling segmentation, we had approximately 700 features (around 7 sets of ~100 features each), while whole tumor segmentation yielded about 100 features. Initially, we used a stricter correlation threshold of 0.9 to remove highly correlated features. However, applying this threshold to the IBSI RF analyses sometimes resulted in retaining very few features (sometimes <10), which could overly limit the feature set and reduce model flexibility. Therefore, we relaxed the threshold slightly to 0.8 in these cases to retain a more balanced number of features, allowing the model more informative inputs for the subsequent machine learning analysis.

For RF with filters, the number of features increased dramatically to thousands (up to >7000 with tumor sampling segmentation). Given this high dimensionality, we applied a stricter threshold of 0.9 to effectively reduce redundancy and computational burden.

In summary, different thresholds were chosen pragmatically based on the total number of features and segmentation strategy, while balancing feature redundancy reduction with preserving sufficient information for model training. Explanations of the threshold selections were added to the Machine Learning Method section.

Discussion:

Comments 6: Could the authors discuss how these models might perform with multimodal imaging (adding T2-weighted or diffusion sequences), which is increasingly relevant in osteosarcoma staging?

Response 6: Thank you for highlighting this aspect. While T2-weighted imaging data is available for many patients in our cohort, due to limited radiologist availability for segmentation, this initial study focused on T1 post-contrast sequences in coronal planes. We recognize that incorporating additional imaging modalities, such as T2-weighted or diffusion-weighted sequences, could provide complementary information and potentially improve model performance. As a next step, we plan to extend our analysis to include multimodal imaging data and different imaging planes, as acknowledged in the discussion section. This will allow us to evaluate whether combining features across sequences enhances the prediction of clinical outcomes in pediatric osteosarcoma.

Comments 7: It would also strengthen the paper to compare directly to previous pediatric OS radiomics studies on therapy response or survival and more explicitly state what is novel beyond adding multi-region segmentation.

Response 7: Thank you for asking. In the introduction, we have mentioned four key areas of novelty 1) there is a lack of standardization in RF extraction and applications in OS studies; 2) studies have generally examined each outcome separately without considering the interdependencies between them; 3) most prior studies have relied on whole-tumor segmentation without considering regional variations within the tumor; 4) there is limited focus on pediatric populations, even though pediatric OS differs biologically and clinically from adult cases. In the Discussion, we have expanded the comparison with previous studies. For example, one multicenter study using T1 post-contrast MRI for predicting chemotherapy response reported an AUC of 0.882. Another study using T2-weighted MRI reported an AUC of 0.708. For survival prediction, diffusion-weighted MRI and T2-weighted MRI studies have reported concordance indices ranging from 0.741 to 0.813. We have revised the discussion to include these points more explicitly and appreciate the reviewer’s suggestion, which helped us improve the clarity and impact of our manuscript.

Round 2

Reviewer 2 Report

Comments and Suggestions for Authors

I have reviewed the re-submission and the authors have carefully amended their manuscript following the additional reviewers' suggestions.